

# FastProtein—an automated software for *in silico* proteomic analysis

Renato Simões Moreira[1,2], Vilmar Benetti Filho[2], Guilherme Augusto Maia[2], Tatiany Aparecida Teixeira Soratto[2], Eric Kazuo Kawagoe[2], Bruna Caroline Russi[1], Luiz Cláudio Miletti[3] and Glauber Wagner[2]

[1] Instituto Federal de Santa Catarina, Gaspar, Santa Catarina, Brazil
[2] Departamento de Microbiologia, Parasitologia e Imunologia, Universidade Federal de Santa Catarina, Florianópolis, Santa Catarina, Brazil
[3] Centro de Ciências Agroveterinárias, Universidade do Estado de Santa Catarina, Lages, Santa Catarina, Brazil

## ABSTRACT

Although various tools provide proteomic information, each tool has limitations related to execution platforms, libraries, versions, and data output format. Integrating data generated from different software is a laborious process that can prolong analysis time. Here, we present FastProtein, a protein analysis pipeline that is user-friendly, easily installable, and outputs important information about subcellular location, transmembrane domains, signal peptide, molecular weight, isoelectric point, hydropathy, aromaticity, gene ontology, endoplasmic reticulum retention domains, and N-glycosylation domains. It also helps determine the presence of glycosylphosphatidylinositol and obtain functional information from InterProScan, PANTHER, Pfam, and alignment-based annotation searches. FastProtein provides the scientific community with an easy-to-use computational tool for proteomic data analysis. It is applicable to both small datasets and proteome-wide studies. It can be used through the command line interface mode or a web interface installed on a local server. FastProtein significantly enhances proteomics analysis workflows by producing multiple results in a single-step process, thereby streamlining and accelerating the overall analysis. The software is open-source and freely available. Installation and execution instructions, as well as the source code and test files generated for tool validation, are available at https://github.com/bioinformatics-ufsc/FastProtein.

## INTRODUCTION

The complexity of high-throughput sequencing data and the need for reproducible analysis are challenges that require integrated workflows (*Wratten, Wilm & Göke, 2021*). Some workflow managers for bioinformatics include community-driven projects and workflow management systems. Galaxy (*The Galaxy Community et al., 2024*) and nf-core (*Ewels et al., 2020*) are examples of community-driven projects, while Nextflow (*Di Tommaso et al., 2017*) and Snakemake (*Mölder et al., 2021*) are script-based workflow management systems. Workflow managers and automated software facilitate reproducible and scalable data analysis. However, Galaxy is the only platform with a user-friendly interface and

Corresponding author
Glauber Wagner,
glauber.wagner@ufsc.br

a point-and-click feature to create workflows (*Wratten, Wilm & Göke, 2021*), while the remaining tools require the command-line interface.

Downstream analysis results in qualitative and quantitative features of proteins, which typically involve using several bioinformatics software packages in tandem but in a non-integrated workflow (*Chen et al., 2020*; *Jiménez-Munguía et al., 2018*). Although Blast2GO (*Conesa et al., 2005*) is an alternative tool widely used for functional annotation, it is closed sourced and requires a license. Proteomic analysis generates a considerable amount of computational data that require bioinformatics analysis (*Vaudel et al., 2016*).

FastProtein is an automated, user-friendly, and publicly available software that integrates functional annotation, database similarity search, and protein feature prediction to enable global proteomic profiling. Furthermore, the *in silico* results obtained through FastProtein can be used to characterize proteins of interest in search of biological insights.

# MATERIALS & METHODS

## Workflow

FastProtein uses a protein FASTA file as input to generate protein profiles. The workflow analysis begins by parsing and standardizing the input for different software (Fig. 1). Parsing and input validation are executed by BioJava (*Lafita et al., 2019*). Sequences with undetermined amino acids (represented by the letter "X") are invalid and removed from the initial dataset before execution.

The first step of FastProtein involves biochemical feature prediction, which provides attributes such as protein length, molecular mass (kDa), hydropathy, isoelectric point (PI), and aromaticity (*Lobry & Gautier, 1994*). These processes are managed and executed through BioJava (*Lafita et al., 2019*). Subsequently, FastProtein identifies the N-glycosylation and endoplasmic reticulum retention domains using the PROSITE (*Sigrist et al., 2013*) database entries PS00001 and PS00014, respectively.

WoLF PSORT (*Horton et al., 2007*) is used to predict the subcellular locations of eu-karyotic organisms. Transmembrane site, signal peptide, and glycosylphosphatidylinositol (GPI)-anchored predictions are performed using TMHMM-2 (*Möller, Croning & Apweiler, 2001*), SignalP 5.0, (*Almagro Armenteros et al., 2019*), and PredGPI (*Pierleoni, Martelli & Casadio, 2008*), respectively. Additionally, the transmembrane domain and signal peptide are predicted using Phobius (*Käll, Krogh & Sonnhammer, 2004*).

Functional annotations are optional and performed using InterProScan (*Jones et al., 2014*). The outputs are merged and parsed to obtain Pfam (*Mistry et al., 2021*) and PANTHER (*Thomas et al., 2022*) domains, InterPro (IPR) annotations (*Jones et al., 2014*), and gene ontology (GO) terms (*Ashburner et al., 2000*). The sets of GO terms associated with the protein are determined by analyzing all databases using InterProScan. These terms are organized into a file that can be imported into the WEGO 2.0 (*Ye et al., 2018*) platform for complementary analysis. This platform is used to group, visualize, compare, and generate GO plots.

GO terms provide quantitative reports on molecular functions, cellular components, and biological processes. This process generates a file containing the GO terms (one line per protein, followed by GO terms in table-separated files).

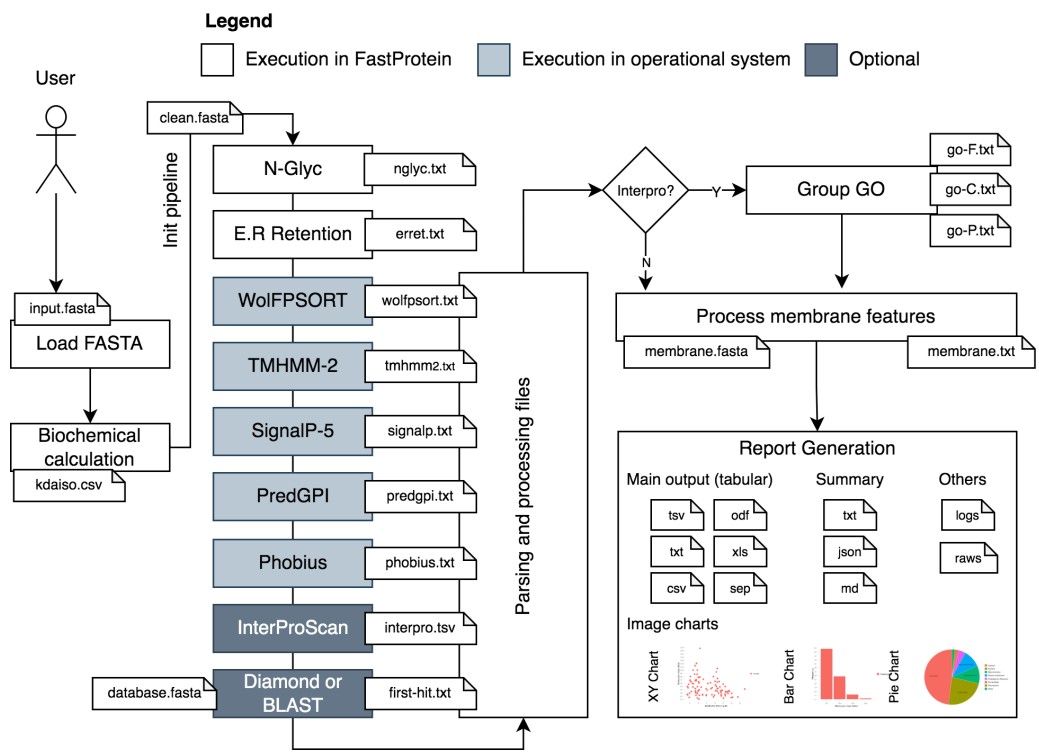

**Figure 1  FastProtein workflow.** A user-provided FASTA file are processed and submitted to third-party software that genarate outputs with GO terms, membrane evidence and other relevant graphs and files.

Another important step in this workflow is similarity analysis. FastProtein returns the best hit for each protein (with its identity and coverage percentage) using the BLASTp (*Camacho et al., 2009*) or DIAMOND (ultra-sensitive mode) (*Buchfink, Reuter & Drost, 2021*) algorithms. Local alignment using BLASTp is only available through the command-line interface (CLI). DIAMOND is the only local aligner available through the web server.

FastProtein provides six features to predict membrane protein: GPI-anchored, two predictions for transmembrane domains (TM, predicted *via* TMHMM-2.0; and PHOBIUS_TM, predicted *via* Phobius), subcellular localization predicted *via* WoLF PSORT (SL), GO, and IPR annotations. The presence of any one of these six features is sufficient to classify a protein as a membrane protein. Finally, a report file and a FASTA file are generated for proteins with membrane-related evidence.

## Output files

FastProtein generates multiple outputs, including both quantitative and qualitative results, as well as FASTA files (Supplemental Information 1). Additionally, it produces an integrated histogram and scatter plot of molecular masses and isoelectric points, along with a bar chart depicting predicted subcellular localizations. The images are created at 300 DPI using Matplotlib (*Hunter, 2007*) and seaborn (*Waskom, 2021*).

Individual protein information is provided in tab-separated values (TSV), comma-separated values (CSV), plain text (TXT), XLS (Microsoft Excel), open document format

**Table 1  Third-party software used by FastProtein.**

| Software/Version | Purpose | Reference |
|---|---|---|
| WoLF PSORT (0.1) | Subcellular location (for eukaryotes only) | *Horton et al. (2007)* |
| TMHMM (2.0c) | Transmembrane predictions of domain sites | *Möller, Croning & Apweiler (2001)* |
| SignalP (5.0b) | Signal peptide prediction and location of cleavage sites in protein | *Almagro Armenteros et al. (2019)* |
| InterProScan (5.61–93.0) | Functional predictions (Ontology terms) | *Jones et al. (2014)* |
| BioJava (7.0.0) | Bioinformatic support library | *Lafita et al. (2019)* |
| Phobius (1.01) | A combined transmembrane topology and signal peptide predictor | *Käll, Krogh & Sonnhammer (2004)* |
| PredGPI (202001) | GPI-Anchor Predictor | *Pierleoni, Martelli & Casadio (2008)* |
| BLASTp (2.10.0) | Sequence aligner for proteins | *Camacho et al. (2009)* |
| DIAMOND (2.0.7) | Sequence aligner for proteins | *Buchfink, Reuter & Drost (2021)* |
| Seaborn (0.13.2) | Python data visualization library | *Waskom (2021)* |
| Matplotlib (3.9.2) | Python library for creating static, animated, and interactive visualizations | *Hunter (2007)* |

(ODF), and separated (SEP) file formats. SEP is a custom format similar to the ProtComp v9 (http://www.softberry.com/) output.

All generated files are stored in a temporary directory within the FastProtein installation directory, named using a universally unique identifier (UUID) created at the start of the run, which enables parallel execution. Upon completion, the temporary directory is renamed to the user-selected output directory (with 'fastprotein _results' as the default). In case of processing errors, the previously generated files can be reused by employing the <-cdt directory>command, which specifies the directory from which the files should be retrieved. This option is only available through the CLI mode. An execution log is saved in the output directory, and the logging level can be set in the CLI mode. Furthermore, the FastProtein Docker container functions as a comprehensive bioinformatics suite, featuring several pre-installed software packages (Table 1) that can be executed independently.

## User friendly web-based interface

A web module was developed using Python (v3.9.2) and Flask (v2.2.3) to execute FastProtein within a Docker container. This interface enables users submit new FastProtein executions and monitor the progress of their tasks (Fig. 2) and visualize results through charts (Fig. 3A) and an interactive table (Fig. 3B). Additionally, it includes modules for managing databases, users, and permissions.

## Computational infrastructure

A Debian-based Docker image (*Merkel, 2014*) is available at https://hub.docker.com/r/bioinfoufsc/fastprotein. This image is 900 MB (compressed) and includes an installation script for InterProScan, which is required for functional annotation (recommended). The third-party software and dependencies used are listed in Table 1 and the commands for each third-party software are detailed in Supplemental Information 2.

The installation guide, usage instructions, and the source code are available at https://github.com/bioinformatics-ufsc/FastProtein. FastProtein can be executed in two different ways: a web-based GUI and through the CLI from a local FastProtein Docker

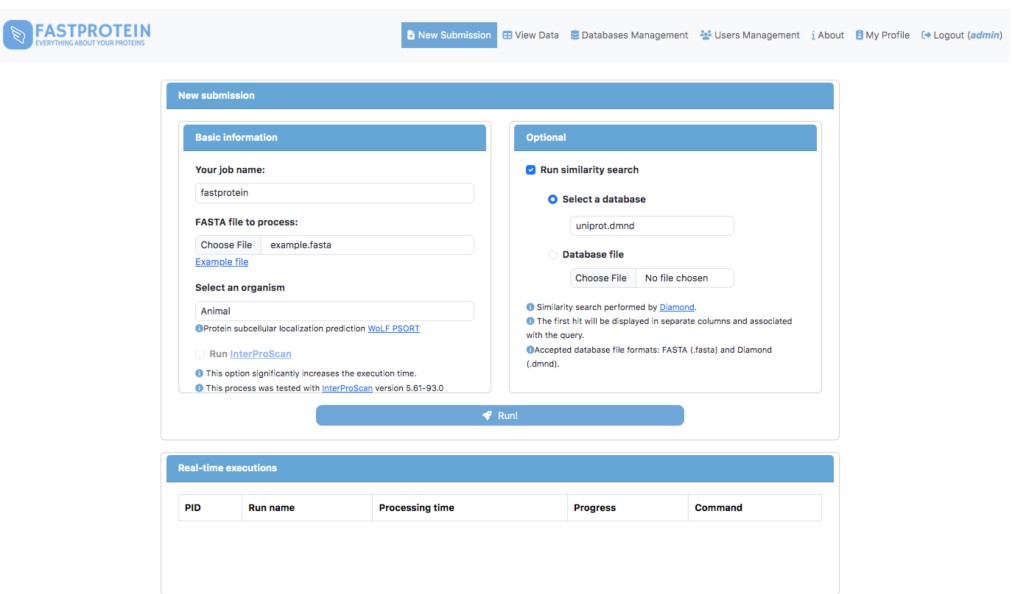

**Figure 2** Main GUI of the FastProtein web module deployed on a local server.

container Users can also build a new FastProtein Docker image using the Dockerfile available at https://github.com/bioinformatics-ufsc/FastProtein.

## Experiments

We used a subset of 125 proteins from the *Plasmodium malariae* proteome (UP000219813) to demonstrate the FastProtein workflow. This test run was performed through the CLI mode using DIAMOND (ultra-sensitive mode) and BLASTp for local alignment.

For proteome-wide analysis and performance benchmarking, we used the following proteomes (strain, Proteome ID) downloaded on April 2, 2023, from UniProt (*The UniProt Consortium et al., 2021*): *Plasmodium vivax* (strain Salvador I, UP000008333), *Trypanosoma brucei* (strain 927/4 GUTat10.1, UP000008524), *Cryptosporidium muris* (strain RN66, UP000001460), *Toxoplasma gondii* (strain ATCC 50861/VEG, UP000002226), *Aspergillus novofumigatus* (strain IBT 16806, UP000234474), and *Cyanidioschyzon merolae* (strain NIES-3377/10D, UP000007014). All proteomes were analyzed using DIAMOND and BLASTp (against the respective proteomes), and each proteome was run in triplicate. The WoLF PSORT dataset was set up to consider the closest organism, as the models are restricted to animals, plants, and fungi.

## RESULTS

Only 124 proteins from the initial dataset were deemed eligible for analysis. Protein A0A1A8WBL6 had invalid sequences (represented by the letter "X") and was removed from the dataset. The total analysis time for this dataset, using DIAMOND, was 2 min and 58 s, with 5 s exclusively required by FastProtein, and the remaining time by third-party software. The total execution time was 7 min and 13 s, using BLASTp, with 3 min and
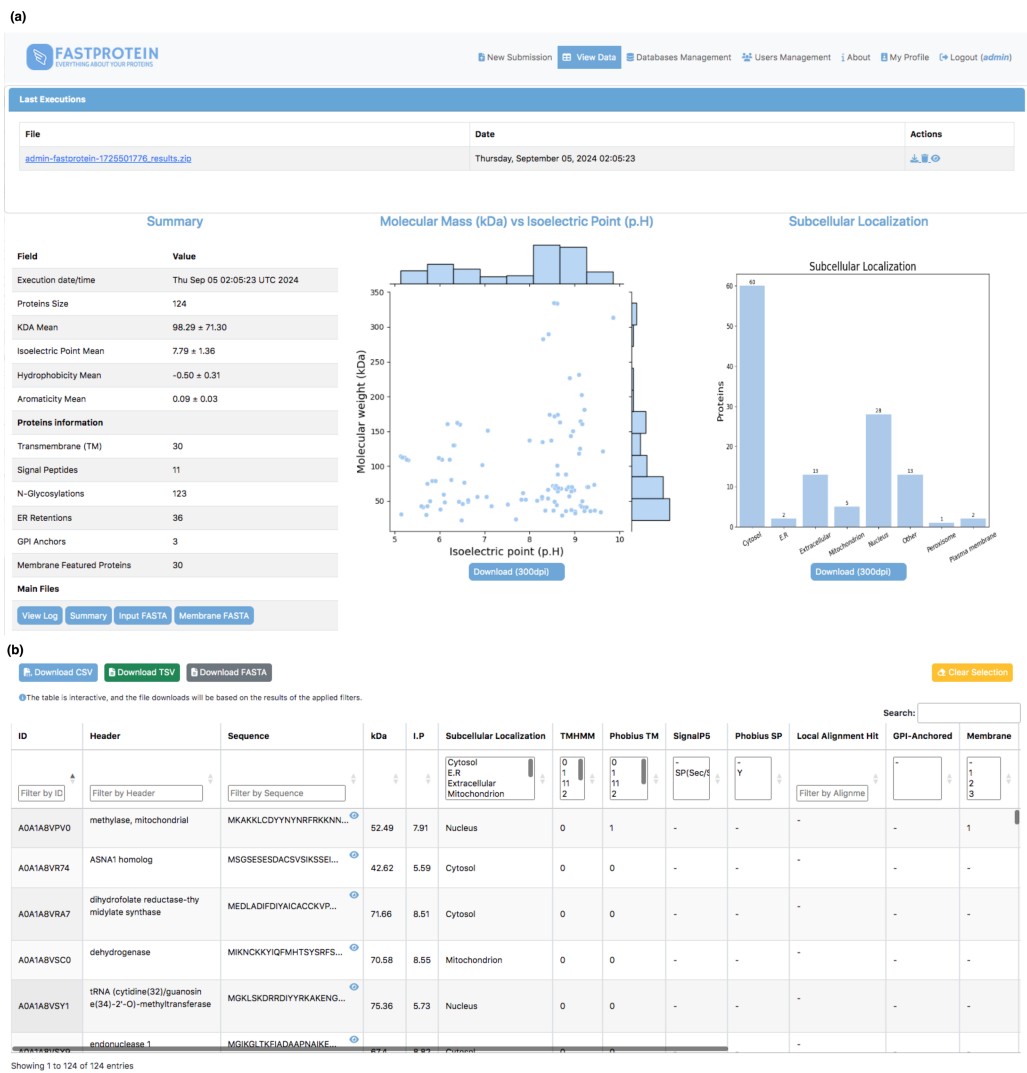

**Figure 3** **Web interface for visualizing data generated by FastProtein.** (A) Executions are listed, and by clicking, the user is directed to the main view, which displays overall information about the run, including scatter plots of molecular weight *vs.* isoelectric point, as well as a bar chart for subcellular localization. (B) An interactive table where the user can filter data according to their needs, with the option to download the filtered data in CSV, TSV, or FASTA format.

49 s used for local alignment, 5 s by FastProtein, and the remaining by other third-party software.

The average molecular weight and isoelectric point in the *P. malariae* dataset were $98.29 \pm 71.30$ kDa and $7.79 \pm 1.36$, respectively. The distribution generated by FastProtein is shown in Fig. 3A. The average hydrophilicity and aromaticity were $-0.50 \pm 0.31$ and $0.09 \pm 0.03$, respectively. Out of the analyzed proteins, 30 were predicted to contain transmembrane domains, three exhibited GPI anchoring, and an additional 30 were estimated to be membrane proteins. Furthermore, among the latter group, four exhibited multiple pieces of evidence supporting their localization to the membrane (GO, IPR,

PHOBIUS_TM, and TM). Among the 124 *P. malariae* proteins, 11 were predicted to have signal peptides and 36 had ER retention domains, with the KEEL and KNEL domains being the most frequently occurring, existing in seven proteins each. Of the 124 proteins, 123 had N-glycosylation domains, with the NNS domain occurring most frequently (116 proteins). The subcellular locations are shown in Fig. 3A, wherein cytosol and nucleus are the most frequent ones, with 60 and 28 proteins, respectively.

For the proteome-wide analysis, the fastest runtime was $28.85 \pm 5.74$ min for the *C. muris* dataset (3,930 proteins; 3,924 processed and six ignored) using DIAMOND. The slowest runtime was $262.35 \pm 2.63$ min for the *T. gondii* dataset (8,404 proteins) using BLASTp. The execution time decreased to $77.75 \pm 8.90$ min with DIAMOND as the local aligner. Even though *A. novofumigatus* had 3,076 more proteins than *T. gondii*, it took $97.42 \pm 0.21$ min to analyze it using BLASTp, and $67.71 \pm 1.73$ min using DIAMOND. The best result in terms of proteins analyzed per minute was obtained for the *A. novofumigatus* dataset with 170 proteins (using DIAMOND), whereas the worst was obtained for *T. gondii* with 32 proteins (using BLASTp). For all DIAMOND executions, the alignment was completed within a few seconds. For BLASTp, the fastest execution required $8.02 \pm 0.05$ min and the slowest required $193.06 \pm 0.92$ min. Considering our entire workflow, the local alignment was the only step with different execution times, which is considerably similar to previously reported results using different alignment algorithms (*Hernández-Salmerón & Moreno-Hagelsieb, 2020*).

The average execution times for each software used in the pipeline were as follows: WoLF PSORT (~2 min), TMHMM-2.0 (~8 min), SignalP5 (~2 min), PredGPI (~2 min), Phobius (~11 min), InterProScan (~25 min), DIAMOND (<1 min), and BLASTp (~52 min). The average time required for the internal execution of FastProtein, file generation, conversions, and calculations were approximately 1 min. Both files generated from the subset of *P. malariae* proteins and proteome-wide rounds are available at https://github.com/bioinformatics-ufsc/FastProtein (including the intermediate files, which were removed during processing). All data analyses are presented in Supplemental Information 3.

## DISCUSSION

FastProtein is a user-friendly and easy-to-install protein analysis pipeline tool that provides important information about protein datasets. FastProtein integrates calculations of molecular weight, isoelectric point, hydropathy, and aromaticity with predictions of subcellular location, transmembrane domains, signal peptide and GPI-anchor, GO, endoplasmic reticulum retention, and N-glycosylation domains. It also integrates results from InterProScan, PANTHER, Pfam, and alignment-based annotation searches. Additionally, the software provides a dataset of proteins with evidence of membrane localization, which is important for immunogenicity studies during vaccine development (*Cheng et al., 2021*; *Kis et al., 2018*) and diagnostic tests, such as ELISA (*De Haro-Cruz et al., 2019*; *Iha et al., 2022*) and western blotting (*Begum, Murugesan & Tangutur, 2022*; *Crescitelli, Lässer & Lötvall, 2021*; *Springhorn & Hoppe, 2019*).

FastProtein outputs files in formats that are widely used in the scientific community, including TSV, XLS, and ODF, as well as high-quality 300 DPI images, which is a widely used standard.

The total execution time of FastProtein depends on the InterProScan functional analysis and the local alignment method. By default, DIAMOND was selected due to its relatively faster execution time, although BLASTp is also available. The global median time for proteome-wide analyses was approximately 116 proteins per minute. This was increased to approximately 142 proteins per minute using DIAMOND, and approximately 90 proteins per minute. Thus, proteomic data from a large dataset can be quickly obtained using FastProtein. Considering differences in execution time and sensitivity, DIAMOND was chosen as the local aligner for the web server. DIAMOND can significantly decrease the alignment time (*Buchfink, Reuter & Drost, 2021*).

The only requirement for using the FastProtein software is the installation of Docker for local runs. FastProtein is an easy-to-use and viable tool for researchers with no background in bioinformatics because it provides a user-friendly interface similar to well-established software such as Blast2GO (*Conesa et al., 2005*), MEGA11 (*Tamura, Stecher & Kumar, 2021*), and MaxQuant (*Prianichnikov et al., 2020*). It also contributes to the initiatives that aim to democratize access to bioinformatics, such as the BioLib (https://biolib.com) and Galaxy (*The Galaxy Community et al., 2024*) projects.

## CONCLUSIONS

FastProtein is a novel and user-friendly pipeline tool for proteomic data analysis that is available for small datasets and proteome-wide studies. Furthermore, it can be used through the CLI mode or a web interface. FastProtein accelerates proteomics analysis routines by generating multiple results in a one-step run. One of the limitations of FastProtein is that it does not yet integrate mass spectrometry data. However, the integration of both raw MS/MS data and data from other protein identification software through mass spectrometry is currently being implemented. The software is open-source and available at https://github.com/bioinformatics-ufsc/FastProtein, along with installation and execution instructions and test files generated for validation.

## ACKNOWLEDGEMENTS

We are thankful to SeTIC (Superintendência de Governança Eletrônica e Tecnologia da Informação e Comunicação) team from the Universidade Federal de Santa Catarina (UFSC) for the all computational infrastructure support and for hosting the FastProtein website.

### Funding

This work was supported by Santa Catarina Research Foundation (Fundação de Amparo à Pesquisa e Inovação of Santa Catarina, FAPESC, Santa Catarina, Brazil) and CAPES (Coordination for the Improvement of Higher Education Personnel, Brazil, Grant: 88881.311316/2018-01). Eric Kazuo Kawagoe, Guilherme Augusto Maia, and Vilmar Benetti Filho received scholarships from CAPES (Coordination for the Improvement of Higher Education Personnel, Brazil). Tatiany AT Soratto was a recipient of a scholarship from Santa Catarina Research Foundation (Fundação de Amparo à Pesquisa e Inovação of Santa Catarina, FAPESC, Santa Catarina, Brazil). The funders had no role in study design, data collection and analysis, decision to publish, or preparation of the manuscript.

### Grant Disclosures

The following grant information was disclosed by the authors:
Santa Catarina Research Foundation (Fundação de Amparo à Pesquisa e Inovação of Santa Catarina, FAPESC, Santa Catarina, Brazil) CAPES (Coordination for the Improvement of Higher Education Personnel, Brazil, Grant: 88881.311316/2018-01).

### Competing Interests

The authors declare there are no competing interests.

### Author Contributions

- Renato Simões Moreira conceived and designed the experiments, performed the experiments, analyzed the data, authored or reviewed drafts of the article, and approved the final draft.
- Vilmar Benetti Filho performed the experiments, analyzed the data, prepared figures and/or tables, and approved the final draft.
- Guilherme Augusto Maia performed the experiments, analyzed the data, prepared figures and/or tables, and approved the final draft.
- Tatiany Aparecida Teixeira Soratto performed the experiments, prepared figures and/or tables, and approved the final draft.
- Eric Kazuo Kawagoe performed the experiments, prepared figures and/or tables, and approved the final draft.
- Bruna Caroline Russi analyzed the data, authored or reviewed drafts of the article, and approved the final draft.
- Luiz Cláudio Miletti conceived and designed the experiments, authored or reviewed drafts of the article, and approved the final draft.
- Glauber Wagner conceived and designed the experiments, authored or reviewed drafts of the article, and approved the final draft.

### Data Availability

The installation and execution instructions, the source code and test data generated for tool validation, are available at GitHub and Zenodo:
- https://github.com/bioinformatics-ufsc/FastProtein.
- Renato Simões, Vilmar Benetti Filho, & ekazuo. (2024). bioinformatics-ufsc/FastProtein: FastProtein-v1.0.0 (v1.0.0). Zenodo. https://doi.org/10.5281/zenodo.13838683.

## Supplemental Information

Supplemental information for this article can be found online at http://dx.doi.org/10.7717/peerj.18309#supplemental-information.

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
