# Peer review of "FastProtein—an automated software for in silico proteomic analysis"

_PeerJ, doi:10.7717/peerj.18309_

## Round 0.1 · original submission · Major Revisions

Dear Authors,

After a thorough review of your manuscript by two reviewers, both have concluded that major revisions are necessary before it can be accepted for publication in PeerJ. They have identified several areas for improvement, including the enhancement of the introduction and the correction of certain sections of the English language.

Reviewer 1 has raised specific concerns about the webserver component. He was unable to test the application, and noted that the webserver lacks visualization features and only offers result downloads in .zip format. Addressing these issues, along with the other observations made by the reviewers, will greatly enhance the quality and usability of your manuscript and application.

Best regards,

Armando Sunny

Reviewer 1 ·

Basic reporting

Moreira et al. present FastProtein a webserver and CLI tool to cohesively run a set of popular protein sequence analysis methods on a user given set of proteins. Analysis of unknown proteins has become a common task for bioinformatics practitioners who might not have deep technical expertise, so easy-to-use software for sequence analysis is timely and useful. I was able to run the docker-based webserver (without alignment stage; see below) and generate the expected output for a FASTA file I supplied. I could not test the webserver, since it does not seem to be currently available. I did not try the CLI interface separately for the Docker container.

The manuscript is clearly written, though perhaps could be much more concise, especially since FastProtein does not introduce any new analyses or algorithms. In general, I worry about longer term maintenance of FastProtein, since it does not use community standards (i.e., Bioconda, Biocontainers, etc.) for installation of itself or its dependencies and only relies on Docker Hub for distribution. The latter is known for deleting containers that are not regularly used, threatening future reproducibility.

Major:
- https://biolib.com/UFSC/FastProtein does not work. Therefore, I cannot test the webserver component.
- Database selection as a FASTA file upload looks to be very inefficient. Even a small reference database, such as the Swiss-Prot, is hundreds of megabytes large, and typical reference databases take up multiple tens of gigabytes in size. There should be some mechanism to prepare databases and then select a prepared database.
- I could not run FastProtein through the Docker-based webserver with FASTA file for database input. Without the FASTA file it works. There is no error message or any other indication of what is going wrong. One recommendation here would be to not daemonize the flask process, so at least some indication of what might be going on is available through the Docker stdout.
- I strongly recommend against packing dependencies as .zip files into git. This will become a heavy maintenance burden. Instead, I would recommend using Conda within Docker to install all dependencies. Additionally, I would also recommend making FastProtein available through (bio)conda.

Minor:
- I was disappointed to see that the webserver contains no visualization features, and only offers result .zip download.
- /bioinformatic/fastprotein/temp should be a Docker volume and all intermediate outputs should be written to it. As far as I can tell, user inputs are placed in /bioinformatic/fastprotein/*.fasta instead of the corresponding temp directory. This will likely lead to errors when multiple users try to use the webserver at the same time. Please check if this is the case.
- The experiments show very little benefit of continued support of blastp. I would recommend only focusing on Diamond for a streamlined user experience and smaller container sizes. Additionally, Diamond in its ultra sensitive mode, should be more sensitive than blastp. From the webserver output, it seems that Blast is used without an option to switch to Diamond.
- Introduction should reference community analysis repositories such as Galaxy, nf-core and workflow managers, such as Snakemake, Nextflow and CWL.
- Please add the version used in the publication and command line call executed (where possible) for each method to Table 1.
- I recommend revising the abstract for conciseness, especially I would focus on what a user can achieve with FastProtein and not how.
- Fig 1b is very confusing. I am not sure it adds a lot of value to the manuscript. I would recommend dropping it without replacement and instead ensure that every label in Fig 1a is clearly readable.
- I recommend adding a non-root user to the Docker container for added reliability / as a best practice (see e.g., https://code.visualstudio.com/remote/advancedcontainers/add-nonroot-user).

Experimental design

See above

Validity of the findings

See above

·

Basic reporting

see below

Experimental design

see below

Validity of the findings

see below

Additional comments

The title “FastProtein – An automated software for in silico proteomic analysis” is quite interesting and well written. I will recommend this paper for publication after major revisions. My comments are given below:


1: It also conducts similarity analysis using BLASTp or DIAMOND algorithms and provides membrane protein evidence with six criteria. Please explain why it not conducts similarity analysis using BLASTp and DIAMOND?

2: Please carefully check the English grammar and spelling mistakes e.g. analyses and analysis.

3: I’ll recommend the authors to please improve the introduction section by adding more data.

4: I’ll recommend the authors to explain this excluding process briefly “From the first dataset comprising 125 proteins, only 124 were deemed eligible for analysis, with protein A0A1A8WBL6 excluded due to the presence of X in its aminoacidic sequence”. Why authors exclude this protein on the basis of X.

5: What are the limitations of FastProtein? Especially when applied on large datasets. Please explain briefly.
6: I’ll also suggest the authors to improve the Table 1.

---

## Round 0.2 · accepted · Accept

Dear Author's,

We are pleased to inform you that your manuscript, “FastProtein – An automated software for in silico proteomic analysis”, has been accepted for publication in PeerJ. We commend you and your co-authors on the effort and dedication in developing FastProtein, a highly accessible and much-needed tool for protein bioinformatics.

Thank you for your contribution to the bioinformatics community. We look forward to seeing your work published and the impact it will have on the field.

Sincerely,

Armando Sunny

Reviewer 1 ·

Basic reporting

Developing accessible software for protein bioinformatics is a timely and important endeavor. For this purpose, Moreira et al. have developed an integrated CLI and webserver called FastProtein to automate many analyses steps that would otherwise be time-consuming to setup and require some technical expertise with bioinformatics software.

Since my last review, the authors have incorporated the majority of my feedback and introduced major improvements to the webserver and software in general. I tried the webserver briefly after I initially received the rebuttal and was impressed by the degree of improvement. However, at the time of writing the webserver seems to down. Running the docker container locally worked without issues.

Minor:
* Line 671, missing space "BLASTp,and"
* Line 850, missing period "implementedThe"
* Line 1086: genarate

* The webserver table header and body become misaligned on ultra-wide screens (see attached screenshot).
* The numeric table filters (TMHMM, Phobius TM, ...) are confusing (in part due to the lexicographic ordering) and difficult to select

Not relevant for publication:
* I still recommend to not bundle the third-party software as zip files within the git repository and instead use community standards such as bioconda or some other package management for installation of third-party software.
* The docker image sizes can likely be reduced slightly by combining some of the RUN commands (e.g. for the default diamond swissprot db)

Experimental design

See above

Validity of the findings

See above

Additional comments

See above

Annotated reviews are not available for download in order to protect the identity of reviewers who chose to remain anonymous.